# Novel Gold Nanorods@Thiolated Pectin on the Killing of HeLa Cells by Photothermal Ablation

**DOI:** 10.3390/pharmaceutics15112571

**Published:** 2023-11-02

**Authors:** Osvaldo Beltran, Mariangel Luna, Marisol Gastelum, Alba Costa-Santos, Adriana Cambón, Pablo Taboada, Marco A. López-Mata, Antonio Topete, Josue Juarez

**Affiliations:** 1Posgrado en Nanotecnología, Departamento de Física, Universidad de Sonora, Unidad Centro, Hermosillo 83000, Sonora, Mexico; a213210001@unison.mx (O.B.); a213207057@unison.mx (M.L.); a213210582@unison.mx (M.G.); 2Grupo de Física de Coloides y Polímeros, Área de Materia Condensada, Departamento de Física de Partículas, Facultad de Física, Universidad de Santiago de Compostela, 15782 Santiago de Compostela, Spain; albacosta.santos@usc.es (A.C.-S.); adriana.cambon.freire@usc.es (A.C.); pablo.taboada@usc.es (P.T.); 3Instituto de Materiales (IMATUS), Universidad de Santiago de Compostela, 15782 Santiago de Compostela, Spain; 4Departamento de Ciencias de la Salud, Universidad de Sonora, Campus Cajeme, Blvd. Bordo Nuevo s/n, Antiguo Providencia, Ciudad Obregón 85040, Sonora, Mexico; marco.lopezmata@unison.mx; 5Departamento de Fisiología, Centro Universitario de Ciencias de la Salud, Universidad de Guadalajara, Sierra Mojada 950, Guadalajara 44340, Jalisco, Mexico; antonio.topete@cucs.udg.mx; 6Departamento de Física, Universidad de Sonora, Unidad Centro, Hermosillo 83000, Sonora, Mexico

**Keywords:** gold nanorods, thiolated pectin, cancer, photothermal conversion, photothermal therapy

## Abstract

Gold nanorods (AuNRs) have attracted attention in the field of biomedicine, particularly for their potential as photothermal agents capable of killing tumor cells by photothermic ablation. In this study, the synthesis of novel AuNRs stabilized with thiolated pectin (AuNR@SH-PEC) is reported. To achieve this, thiolated pectin (SH-PEC) was obtained by chemically binding cysteamine motifs to the pectin backbone. The success of the reaction was ascertained using FTIR-ATR. Subsequently, the SH-PEC was used to coat and stabilize the surface of AuNRs (AuNR@SH-PEC). In this context, different concentrations of SH-PEC (0.25, 0.50, 1.0, 2.0, 4.0, and 8.0 mg/mL) were added to 0.50 mL of AuNRs suspended in CTAB, aiming to determine the experimental conditions under which AuNR@SH-PEC maintains stability. The results show that SH-PEC effectively replaced the CTAB adsorbed on the surface of AuNRs, enhancing the stability of AuNRs without affecting their optical properties. Additionally, scanning electron and atomic force microscopy confirmed that SH-PEC is adsorbed into the surface of the AuNRs. Importantly, the dimension size (60 × 15 nm) and the aspect ratio (4:1) remained consistent with those of AuNRs stabilized with CTAB. Then, the photothermal properties of gold nanorods were evaluated by irradiating the aqueous suspension of AuNR@SH-PEC with a CW laser (808 nm, 1 W). These results showed that photothermal conversion efficiency is similar to the photothermal conversion observed for AuNR-CTAB. Lastly, the cell viability assays confirmed that the SH-PEC coating enhanced the biocompatibility of AuNR@SH-PEC. Most important, the viability cell assays subjected to laser irradiation in the presence of AuNR@SH-PEC showed a decrease in the cell viability relative to the non-irradiated cells. These results suggest that AuNRs stabilized with thiolated pectin can potentially be exploited in the implementation of photothermal therapy.

## 1. Introduction

Cancer stands as a significant global health challenge, primarily attributed to its widespread prevalence, incidence, and high mortality rates. According to the World Health Organization (WHO), in the year 2020, there were 32.74 million individuals afflicted by cancer, with 12.36 million new cases diagnosed, resulting in 6.31 million deaths attributed to this condition in the same year (GLOBOCAN). Traditional cancer treatments encompass surgery, radiotherapy, and chemotherapy. These strategies consist of eradicating the tumor cells and controlling the exacerbated cell growth in the affected tissue or organ and avoiding their dissemination of cancer cells to another body parts [1]. Despite chemotherapeutic agents having shown high efficacy in killing tumor cells, the use of these drugs has been limited due to their low specificity and high cytotoxicity for both healthy cells and cancerous cells, causing severe side effects in the patients. The challenges mentioned can be effectively addressed through the development of innovative functional nanomaterials, enabling the exploration of new therapeutic approaches for combatting cancer. Notable examples include lipid nanoparticles and dendrimers, which offer a precise way to deliver therapeutic agents to cancer cells in target tissues [2], and magnetic and silica nanoparticles proposed for early detection and the application of magnetic hyperthermia therapy [3], while carbon nanotubes have shown potential in photothermal therapies [4]. On the other hand, gold and silver nanoparticles have been used for targeted drug delivery to enhance the efficacy of chemotherapy and phototherapy [5,6,7].

The optical properties of gold nanoparticles (AuNPs) have captured the interest of the biomedical field, as they offer valuable potential for the creation of innovative and promising techniques for imaging, diagnostics, and therapy [8]. With these purposes, AuNPs can be synthesized in diverse shapes, such as spherical, cubic, hollow shell, and cylindrical shapes, among others [9]. A hallmark of these AuNPs is their tunable adsorption bands within the visible and near-infrared of the electromagnetic spectra, known as surface plasmon resonance (SPR). AuNRs exhibit two absorption bands: (i) the transversal surface plasmon resonance (TSPR) and (ii) the longitudinal surface plasmon (LSPR); the first one, occurring at 510–530 nm, results from the collective oscillation of the surface electrons along the transverse axis. In contrast, the second one, appearing at 600–100 nm in the near-infrared (NIR) light range, arises from the collective oscillation of the surface electrons along the longitudinal axis [10]. In this regard, the LSPR has great relevance for biomedical applications [11] due to its strong light absorption of NIR light and conversion of light to heat. Thus, AuNRs have been explored as photothermal agents to implement novelty therapies against cancer, focusing on destroying cancer cells via photothermal ablation [12]. It is important to note that NIR, ranging from 700 to 900 nm of the electromagnetic spectrum, is known as the biological spectral window due to the minimal absorption of electromagnetic radiation by water and tissue components. This feature allows for deep tissue penetration and efficient absorption light by AuNPs, facilitating the development of localized therapies [13]. However, the presence of hexadecyltrimethylammonium bromide (CTAB) used in the synthesis of AuNRs limits its biological applicability [14]. Therefore, the quest for alternative stabilizing agents is of greatest importance. In this regard, stable aqueous solutions of AuNRs have been obtained by several approaches using different materials [15]. Among them, biocompatible synthetic (PEG and PVP) and natural polymers (proteins, polysaccharides), as well as lipids, have also been used to coat and protect gold nanorods [16]. These stabilization techniques enable precise control over the physicochemical attributes of gold nanorods (AuNRs) while augmenting their functionality. This broadens the spectrum of potential applications for AuNRs, encompassing areas such as biomedicine, electronics, and beyond [17,18,19,20,21]. Polysaccharides have emerged as alternatives due to their low cost of production, biocompatibility, and reactivity. Anionic and cationic polysaccharides, such as alginic acid [22], hyaluronic acid [23], chitosan [24], gum Arabic [25] cellulose [26], starch [27] thiolated dextran [28] and PEC [29,30,31] have been used to coat and stabilize spherical gold nanoparticles [32,33,34,35,36]. In the present report, pectin (PEC) has been proposed to stabilize gold nanorods. This biopolymer is an anionic heteropolysaccharide, which, like other biopolymers, has been widely considered in the pharmaceutical area as an encapsulation agent for drugs and active compounds thanks to its solubility, biocompatibility, and biodegradability properties [37]. Additionally, PEC has a grafted structure, distinct from other polysaccharides with a linear structure used in the stabilization of gold nanoparticles and gold nanorods [31,38]. Furthermore, PEC has been the subject of numerous studies and applications since its chemical composition facilitates its functionalization or chemical modification with different types of molecules [30,39,40]. It is important to note that, at least in the consulted bibliography by this research group, there are no reports in which the PEC was used to stabilize AuNRs. In this regard, the principal aim of this research work was to develop an easy and reproducible method to produce a stable and biocompatible AuNR with a promising application as a photothermal agent. In this regard, thiolated pectin (SH-PEC) was used to replace the CTAB (a surfactant commonly used in the synthesis of AuNRs) adsorbed into the surface of AuNRs and coating the surface of the gold nanostructure, providing it stability in an aqueous medium.

## 2. Materials and Methods

### 2.1. Reagents and Materials

The PEC (74% D-glucoronic acid), Cysteamine Hydrochloride, N-(3-Dimethylaminopropyl)-N’-ethylcarbodiimide (EDAC), N-Hydroxysuccinimide (NHS), N, N-Dimethylformamide (DMF), Hexadecyltrimethylammonium bromide (CTAB), Hydrogen -Tetrachloroaurate (III) (HAuCl_4_), Silver Nitrate (AgNO_3_), Ascorbic Acid and Sodium Tetrahydroborate (NaBH_4_) were obtained from Sigma Aldrich, and they were used as received.

### 2.2. Potentiometric Titration

To determine the number of ionizable groups in the PEC molecule, a potentiometric titration was used. Briefly, 100 mg of PEC was dissolved in 100 mL of deionized water, and 10 mL of HCl 0.1 M was added to ensure that all PEC molecules were protonated. The potentiometric titration was performed by adding 0.1 mL of NaOH 0.1 M in intervals of 60 s, keeping the solution at room temperature (25 °C) and under gently magnetic stirring (100 RPM) using a pH meter GLP 22 (Barcelona, Spain) [41].

### 2.3. Thiolated Pectin (SH-PEC) Synthesis

Thiolated pectin (SH-PEC) was synthesized by amidation reaction between the carboxylic acids of PEC with the amino group of cysteamine [42]. Briefly, 1.5 g of PEC was dissolved in 100 mL of water, and then after, 0.132 g of EDC was added and left to react for 2 h. Then, 0.081 g of NHS was added and left to react for another 24 h. The unreacted components were removed by dialysis using Spectrum™ Labs Spectra/Por™ 1 6–8 kD MWCO Standard RC Dry Dialysis Kits membranes. Briefly, bags of 50 mL each were prepared and placed in a solution with 10 mM HCl for 72 h, changing the medium every 12 h. The resulting product was lyophilized and stored at 4 °C.

### 2.4. Infrared Spectroscopy of SH-PEC

To ensure the success of the amidation reaction to get thiolated PEC (SH-PEC), FTIR-ATR spectra (400–4000 cm^−1^) were obtained in ATR mode using a Perkin Elmer Spectrum Two Instrument (Shelton, CT, USA), at a resolution of 2 cm^−1^.

### 2.5. Nuclear Magnetic Resonance (^1^H NMR)

The degree of substitution (*DS*) of PEC by cysteamine was determined using ^1^H NMR spectroscopy. ^1^H NMR spectra was recorded using a Bruker DRX-500 spectrometer (Avance, 400 MHz). For the analysis, both PEC and SH-PEC solutions were prepared at a concentration of 1 mg/mL using D_2_O as a solvent. The NMR spectra were recorded at 300 K, using tetramethyl silane as reference. The *DS* of SH-PEC was calculated by comparing the signal area of the H1′ of cysteamine (3.40 ppm) to the signal area of H4 (4.40 ppm) corresponding to the galacturonic acid residues [43]:$$DS=\frac{{H1}^{\prime }\left(3.40\right)}{2H4\left(4.40\right)}\times 100\%$$

### 2.6. Gold Nanorods Synthesis

AuNRs were synthesized by the seed-mediated method [44]. All glassware were washed with aqua regia to remove any contaminants. On one hand, the gold seeds were prepared by adding 0.5 mL of ice-cold NaBH_4_ (10 mM) to 5.0 mL of an aqueous solution of CTAB (0.1 M) and HAuCl_4_ (0.48 mM). The solution was vigorously stirred for 2 min and then kept undisturbed at 28 °C for 30 min. On the other hand, 100 mL of AuNR growth solution was prepared by mixing CTAB, HAuCl_4_, AgNO_3_, HCl, and ascorbic acid at final concentrations of 0.1 M, 0.5 mM, 0.085 mM, 0.08 M, respectively, and mixed for 5 min at 500 rpm. Thereafter, 100 μL of Au seeds solution were added to the growth solution, the glass bottle was inverted two times, and then, the solution was kept without perturbation. The color of this solution slowly changed to wine-red color over a period of 3 h. Then, the particle solution was centrifuged (15,000 rpm 28 °C) and redispersed in 100 mL of a 10 mM CTAB solution.

### 2.7. SH-PEC Coated AuNRs

Briefly, 10 mL of AuNRs was washed by centrifugation to eliminate the excess of CTAB and was suspended in 1 mL (10% of the initial volume) of water. Then 0.5 mL of AuNRs solution was added to 4.0 mL of SH-PEC solution (0.25, 0.50, 1.0, 2.0, 4.0 and 8.0 mg/mL), and the final mixture was stirred for 4 h. Finally, the SH-PEC-coated AuNRs (AuNR@SH-PEC) were centrifuged three times to ensure the removal of CTAB (15,000 rpm 28 °C for 30 min) and redispersed in 4.5 mL of water.

### 2.8. UV-Vis Absorption Spectroscopy, STEM, AFM, DLS and Zeta Potential Measurements of AuNRs and AuNR@SH-PECs

UV-Vis spectra were obtained by using a spectrophotometer Agilent 8453 scanning from 400 to 1000 nm. The morphological characterization was conducted using both scanning transmission electron microscopy (STEM, FEI “Inspect F-50” instrument) and atomic force microscopy (AFM, JEOL instrument, JSPM 4210, Tokyo, Japan). STEM analyses were prepared applying a drop of the AuNR@SH-PEC onto carbon-coated copper grids and dried overnight. For AFM analysis, a drop of the AuNR@SH-PEC was applied onto a freshly cleaved mica and left to dry; images were obtained in the noncontact mode using silicon nitride cantilevers (NSC15, MicroMash, Billerica, MA, USA). Particle sizes and zeta potential of AuNR@SH-PEC were measured using Zetasizer-Nano Zs (Nanoseries, Malvern Instruments, Worcestershire, UK).

### 2.9. Inductively Coupled Plasma-Mass Spectroscopy (ICP-MS)

The gold concentration in solution was determined by inductively coupling plasma mass spectroscopy (ICP-MS) in a Varian 820-MS equipment (Agilent Technologies, Santa Clara, CA, USA), following a previously reported protocol [45]. Briefly, 1 mL of AuNRs was dissolved into 0.3 mL HCl (37% (*v*/*v*)) and 0.1 mL HNO_3_ (70% (*v*/*v*)). Solutions were diluted with deionized water until reaching a final volume of 2 mL. The intensity of the emission wavelength was measured and compared to a standard solution.

### 2.10. Photothermal Properties of AuNR@SH-PEC

The photothermal properties of AuNR@SH-PECs were analyzed by light irradiation using a continuous wave fiber coupled diode laser source at 808 nm wavelength (1.0 W/cm^2^). The temperature rise was measured using a k type thermocouple connected to a digital thermometer, AMPROBE TMD-51. The samples were irradiated for periods of 30 min of turn-on laser and 30 min of turn-off laser.

The photothermal conversion efficiency (*η*) was determined using a previously described model [46]:(1)$$\eta =\frac{hS\left(T_{eq}-T_{surr}\right)-Q_{0}}{P(1-10^{{-A}_{808}})}$$
where *η* is the photothermal conversion efficiency, *h* is the heat transfer coefficient, *S* is the surface area of the system, *T_surr_* is the temperature of the surrounding environment (25 °C), P is the power output of the laser irradiation (1 W), A_808_ is the absorbance of the AuNR@SH-PEC at 808 nm, and *Q*_0_ represents the heat dissipated by the quartz cuvette water system, which was determined by adjusting the heating of the system without the AuNRs with the help of the following equation:(2)$$∆T\left(t\right)=\tau_{s}\left[\frac{Q_{0}}{\Sigma_{i}\;m_{i}C_{i}}\right](1-e^{-\frac{t}{\tau_{s}}})$$

### 2.11. In Vitro Cell Cytotoxicity and Photothermal Effect

The cytotoxicity of AuNR@SH-PEC was tested in vitro by means of the CCK-8 cytotoxicity assays. Cervical cancer HeLa and fibroblast Balb/c 3t3 cells were seeded into 96-wells plates (1 × 10^4^ cells/well) and grown for 24 h at an optical confluence of 80–90% under standard culture conditions in 100 μL growth medium. After 24 h incubation at 37 °C, 100 μL of AuNRs (1 × 10^11^–6.25 × 10^9^ AuNRs/mL) with different SH-PEC concentrations (1.0–8.0 mg/mL SH-PEC) in the corresponding cell culture medium were injected into the wells and incubated for 24 h. AuNRs coated with CTAB were used as control. After the corresponding incubation, the culture medium was discarded, cells were washed with 10 mM PBS (pH 7.4) three times, and a fresh culture medium (100 μL) containing 10 μL of CCK-8 reagent was added to each well. After 2 h, the absorption of the samples was measured at 450 nm with a UV-Vis microplate absorbance reader (Bio-Rad model 689, Hercules, CA, USA). Cell viability (SR, survival rate) was calculated as follows:$$SR=\frac{Abs\;Sample}{Abs\;Blank}\times 100$$
where Abs Sample is the absorbance at 450 nm for cell samples, and Abs Blank is the absorbance corresponding to the sample controls without the particles. The cytotoxic activity of AuNRs with irradiation was evaluated as mentioned before, but with some differences. The cells were seeded into 96-wells plates (1 × 10^4^ cells/well) and grown for 24 h at an optical confluence of 80–90% under standard culture conditions in 100 μL growth medium. After 24 h incubation, cells were treated with AuNR@SH-PEC at a concentration of 6.25 × 10^9^ AuNRs/mL (which is the concentration at which the AuNR@SH-PEC showed an increase in cell viability). After that, cells were irradiated for 5 min with 808 nm laser. The cytotoxicity was measure after 48 h with the same CCK-8 protocol as before.

### 2.12. ROS Generation

Intracellular ROS generation was determined by the ROS assay kit based on the fluorogenic dye molecule 5(6)-Carboxy-2’,7’-dichlorofluorescein diacetate (DFCA-DA). Cervical cancer HeLa and fibroblasts Balb/c 3t3 cells were seeded into 96-wells plates (1 × 10^4^ cells/well) and grown for 24 h at an optical confluence of 80–90% under standard culture conditions in 100 μL growth medium. After 24 h incubation at 37 °C, 100 μL of AuNRs (6.25 × 10^9^ AuNRs/mL) with different SH-PEC concentrations (1.0–8.0 mg/mL SH-PEC) in the corresponding cell culture medium were injected into the wells and incubated for 24 h. Afterwards, the culture medium was changed for a fresh one, and cells were irradiated with an 808 nm CW laser at 1.0 for 5 min. Afterwards, the ROS production was analyzed using the corresponding kit following the manufacturer’s instructions using a Fluorostar Omega plate reader (BMG Labtech, Ortenberg, Germany). Measurements were made in triplicate.

### 2.13. Statistical Analysis

GraphPad Prism8 (GraphPad Software Inc., San Diego, CA, USA) was used for statistical analysis and graphing. Quantitative data were expressed as the mean ± standard deviation (SD) of three independent experiments (n = 3). All the statistical analyses were assessed using a *t*-Test. A *p*-value of less than 0.05 was considered statistically significant.

## 3. Results and Discussion

### 3.1. Number of Ionizable Groups on PEC

PEC was chemically modified through an amidation reaction, where the carboxilic acids of PEC reacted with the amino groups of cysteamine in a theoretical degree substitution of 10%. Prior to the chemical modification, the number of carboxylic acids present in the PEC was determined by a NaOH titration. Figure 1 shows the potentiometric titration curve recorded for the PEC solution, following variations in pH and conductivity (χ) as the NaOH added to the biopolymer solution increases. The titration curve showed three distinctive zones. The initial zone depicts a slight increase in pH and a decrease in electrical conductivity. This indicates the neutralization of the H^+^ ions previous added to the aqueous medium. Then, the first equivalence point is attained when the excess of H^+^ ions are neutralized [41]. In the second zone, the deprotonation of the carboxylic acids present in the PEC chain takes place. At this point, the pH values progressively increased while the χ value decreased quickly until the deprotonation of the PEC was complete [41]. In the third zone, the pH and χ values change very slowly, indicating that PEC is completely deprotonated and there is an excess of NaOH [41]. The number of ionizable groups on the PEC molecules was determined as in Farris [41]. Based on this, it was found that 4.6 × 10^−^^4^ moles of NaOH were necessary to neutralize the number of -COOH groups present in 0.1 g of PEC. Therefore, the degree of esterification is around to 22% (78% of acid groups), which is close to the data reported by the supplier (≥74% acid groups).

### 3.2. SH-PEC Synthesis

The success of the chemical modification of the PEC molecule with cysteamine was confirmed through FTIR-ATR. Figure 2 shows the FTIR-ATR spectra obtained for PEC (black line) and SH-PEC (yellow line). The PEC spectrum shows that the characteristic bands and peaks inherent to the PEC molecule distinguish a wide band around 3300 cm^−^^1^ attributed to the -OH stretching of carboxylic acid. The peak showed at 1730 cm^−^^1^, corresponding to the stretching bond of the -C=O of the methyl esters of the carbonyl groups, and the peak at 1600 cm^−^^1^ is assigned to the asymmetric tension of -COO^−^. On the other hand, the SH-PEC FTIR-ATR spectrum contrasted significantly from that of PEC. A new peak located at 2300 cm^−^^1^ was assigned to the -SH groups indicating that cysteamine residues have been chemically attached to the PEC [47]. Additionally, the 1600 cm^−^^1^ band in the SH-PEC spectrum shows a small shoulder at 1630 cm^−^^1^, indicating the formation of secondary amide groups, a result of the chemical reaction between the -NH_2_ of the cysteamine and -COOH groups of the cysteamine and PEC [48].

Figure 3 shows the ^1^H NMR spectra recorded for both PEC (black line) and SH-PEC (red line). The ^1^H NMR spectrum for native PEC displays a complex set of signals ranging from 3.78 to 3.65 ppm. Signals corresponding to esterified methoxyl groups and the H_2_ of galacturonic residues appeared at 3.77 and 3.70 ppm, respectively [43]. The double peak observed between 4.45 and 4.35 ppm can be assigned to the H4 of galacturonic acid (4.40 ppm) and esterified galacturonic acid residues (4.46 ppm) [49]. In contrast, the SH-PEC spectrum exhibits a triple signal at 3.40 ppm, which corresponds to the protons of cysteamine (tagged as H1′) [50]. To determine the degree of substitution for PEC, the area ratios of H1′ and H4 (4.40 ppm) [43] was considered, resulting in a *DS* of 6.9%.

### 3.3. Determination of AuNRs Concentration

The concentration of the AuNRs obtained at the end of the synthesis process was determined using inductively coupled plasma mass spectroscopy (ICP-MS). Then, the number of AuNRs was calculated based on the number of gold atoms per AuNR [45]. To ascertain the mass of a single AuNR, the gold density (ρ = 19.3 g/cm^3^) was multiplied by the volume of an AuNR, which was determined by the TEM micrograph analysis (assuming that AuNR adopts cylindrical geometry with an average length of 60 nm and a width of 15 nm). Then, the molar mass was calculated by multiplying the mass of a single AuNR by Avogadro’s number (N_A_ = 6.022 × 10^23^). ICP-MS results indicated a gold concentration of 0.0917 g/L. From this result, the molar concentration of AuNRs determined was 1.12 × 10^−^^9^ M (1.12 nM).

### 3.4. AuNRs Coated with SH-PEC (AuNR@SH-PEC)

The role of SH-PEC as a stabilizer agent for AuNR was assessed by UV-Vis absorption spectroscopy. In this regard, different amounts of SH-PEC (0.25–8.0 mg/mL) were added to an AuNRs solution (1.12 × 10^−^^9^ M) with the aim of determining the optimal concentration of SH-PEC needed to get stable AuNRs. The UV-Vis absorption spectrum in Figure 4 shows the typical spectral features of CTAB-stabilized AuNRs, distinguishing two characteristic absorption bands, one located at a maximum wavelength (λ_max_) of 450 nm corresponding to the TSPR, and a second one located at around 780 nm corresponding to the LSPR. However, spectra recorded for AuNRs stabilized with SH-PEC depicted important changes in TSPR and LSPR at certain SH-PEC concentrations. At 0.25 mg/mL (purple line) and 0.5 mg/mL (dark blue line) of SH-PEC, the absorption intensity diminished for both TSPR and LSPR, and a blue shift in the λ_max_ of LSPR were denoted, suggesting that the structure of AuNR is lost due to the fact that the amount of SH-PEC is too low to cover and stabilize the surface of the nanorods. The AuNR@SH-PEC stabilized with 1.0 mg/mL of SH-PEC (green line) resulted in stability in aqueous media; however, a slight broadening of the LSPR was observed due to the formation of small clusters of AuNR@SH-PEC, suggesting that the SH-PEC amount added to stabilize the AuNR solution was in the threshold to cover the surface of the AuNRs. Interestingly, the absorption spectra recorded for AuNR@SH-PEC stabilized with SH-PEC at concentrations of 2.0 mg/mL (pink line) and 4.0 mg/mL (blue line) showed that the transversal and longitudinal SPRs are localized at the same λ_max_ shown by AuNRs stabilized with CTAB, albeit the absorption intensity diminished due to the biopolymeric coating. Finally, the spectrum recorded for the AuNR@SH-PEC stabilized using 8.0 mg/mL of SH-PEC (red line) shows a decrease in the absorption intensity of both TSPR and LSPR; this could be due to the increased thickness of the polymeric coating on the surface of AuNRs. All these results were corroborated by dynamic light scattering (DLS) and Zeta potential (ζ) experiments.

### 3.5. Dynamic Light Scattering (DLS) and Zeta Potential (ζ) Measurements

DLS and ζ measurements were used to investigate how SH-PEC coating affects the size and stability of the AuNRs (AuNR@SH-PEC). Table 1 shows the hydrodynamic size recorded from DLS for the AuNR@SH-PEC. The size for AuNR@SH-PEC fluctuated depending on the amount of SH-PEC used to stabilize the AuNRs. For instance, with the lowest concentration of SH-PEC (0.25 and 0.50 mg/mL), the sizes of AuNR@SH-PEC were 177 and 104.6 nm, respectively. These results suggest that the surface of the AuNR is partially covered by SH-PEC. This is possibly due to the fact that SH-PEC molecules interact with more than one AuNR, leading to the degradation, aggregation, and eventual precipitation of AuNRs [51]. On the other hand, for AuNRs stabilized with SH-PEC concentrations (1.0–2.0 mg/mL), the sizes of the AuNRs that were stabilized with higher concentrations of SH-PEC (1.0–2.0 mg/mL), and the size enlarged from 50.8 nm to 68.4 nm, respectively, remaining stable. This result suggests that the SH-PEC is adsorbed onto the surface of the AuNRs in a monolayer, maintaining the dimensions of the AuNRs. Finally, the size of AuNR@SH-PEC stabilized with the highest SH-PEC concentration (4.0 and 8.0 mg/mL) was 114.1 and 190.9 nm, respectively. On the basis of these results, it can be concluded that the surface of the AuNRs was completely coated as the amount of the thiolated biopolymer increased, until a multilayer was coated due to the excess of SH-PEC molecules in the aqueous media. These results agree with the results observed by UV-Vis spectroscopy, where at low concentrations of SH-PEC (0.25 and 0.5 mg/mL), a decrease in the absorption intensity and a blue shift of the λ_max_ of longitudinal SPR were observed, suggesting the destabilization of the AuNRs. Meanwhile, at intermediate and higher concentrations of AuNR@SH-PEC (1.0–2.0 mg/mL), the λ_max_ of the longitudinal SPR remains similar to the AuNRs stabilized with CTAB, supporting the role of SH-PEC as stabilizer agent. Based on UV-Vis spectroscopy and the DLS results, it can be concluded that the physical and optical properties of AuNR@SH-PEC with SH-PEC in the range from 1.0 to 8.0 mg/mL can be adequate to be exploited as photothermal devices.

The ζ potential measurements were used to determine the surface charge of the AuNR@SH-PEC and taken to a parameter to confirm (i) the coupling of the SH-PEC molecules in the AuNRs surface and (ii) to get a parameter of stability. Figure 4 shows the ζ potential values of the AuNRs stabilized with CTAB and with different concentrations of SH-PEC (0.25, 0.50, 1.0, 2.0, 4.0 and 8.0 mg/mL). The AuNRs stabilized with CTAB, used as a control, showed a ζ potential of 14 ± 1.8 mV. This is due to the cationic nature of the CTAB molecule. Since PEC is an anionic biopolymer [32], the surface charge recorded for AuNR@SH-PEC resulted in negative values, indicating that the CTAB molecules absorbed on the surface of the gold nanorods were replaced by SH-PEC molecules, as can be observed in Figure 5. Additionally, the ζ potential recorded for the first concentrations of SH-PEC used to stabilize AuNRs, particularly those of 0.50 mg/mL, showed more negative ζ values. These values contrasted with the ζ potential recorded for the following concentrations of SH-PEC (1.0–8.0 mg/mL), since the absolute value of ζ potential progressively decreases from |7.79| to |3.43| mV. This result suggests that the SH-PEC molecules adsorbed on the AuNRs surface formed a monolayer (1.0 and 2.0 mg/mL), as was explained in UV-Vis and DLS results. Additionally, the progressive decrease in the ζ value of AuNR@SH-PEC (4.0 and 8.0 mg/mL) can be related to the screening of the anionic carboxylate groups due to the intermolecular interactions between the SH-PEC molecules, mediated by -COO^−^ and -OH groups present on the chemical structure of SH-PEC, forming a multilayer coating and diminishing the number of -COO^−^ exposed to the surface AuNR@SH-PEC. Based on UV-Vis spectroscopy, DLS and ζ potential results, it can be concluded that the physical and optical properties of AuNR@SH-PEC with SH-PEC in the range from 1.00 to 8.00 mg/mL can be adequate to be exploited as photothermal devices.

### 3.6. Scanning Transmission Electron Microscopy (STEM) and Atomic Force Microscopy (AFM)

Figure 6a,c,e,g shows the STEM and AFM micrographs recorded for AuNR-CTAB and for the AuNRs stabilized with different concentrations of SH-PEC. The STEM images show that the AuNRs have a rod-like shape with longitudinal and transversal axes of around 60 nm and 15 nm, respectively, corresponding to a characteristic aspect ratio of 4:1. On the other hand, due to the low electron density of the SH-PEC, it is difficult to observe the biopolymer adsorbed on the surface of AuNR by the STEM tool; therefore, the AFM technique was used to confirm the presence of SH-PEC adsorbed onto the surface of the AuNR@SH-PEC. Figure 6b,d,f,h shows the AFM images recorded for the AuNR@SH-PECs. AFM images show that the sizes of AuNR@SH-PECs were considerably larger than the AuNR@SH-PEC sizes observed by STEM, suggesting that AuNR@SH-PECs form aggregates because of the drying process. Additionally, it can be observed that the size of the AuNR@SH-PEC aggregates is larger as the SH-PEC amount used in the stabilization process of AuNR increase (211.8 ± 25.4 nm). Interestingly, the AuNR@SH-PEC stabilized with SH-PEC at concentrations of 1.0 mg/mL showed the lowest size, probably due to the fact that the biopolymer coat adsorbed on the surface of the gold nanostructure is conformed by a thinner layer that avoids the formation of large aggregates. Meanwhile, at higher concentrations of SH-PEC, the polymer is adsorbed on the surface of AuNR as a multilayer coat facilitating the interaction between AuNR@SH-PECs and allowing its aggregation during the drying process. Thus, in accordance with STEM and AFM results, it can be concluded that SH-PEC is able to coat the surface of the AuNRs, providing them stability in an aqueous solution without significantly affecting the optical properties of gold nanorods.

### 3.7. Photothermal Assays

To evaluate if the photothermic properties of the AuNRs were affected by the SH-PEC coating, the AuNR@SH-PEC solutions were subjected to three continuous cycles of turn-on and turn-off laser irradiation using a CW 808 nm diode laser (CNI, Changchun, China). The temperature increase was monitored over time. It is important to note that the AuNR@SH-PEC solutions, when subjected to laser irradiation, consistently reached peak temperatures across all three heat cycles. This behavior suggests that the AuNR@SH-PEC remains stable throughout the heat cycles. The SH-PEC coating, in addition to stabilizing the AuNR@SH-PEC solution, also provides thermal stability to the AuNR, preserving its shape. Likewise, the photothermal conversion efficiency was determined following the model described by Roper et al. [46]. In all cases, a fast increase in the temperature of aqueous media was observed until it reached a quasi-equilibrium temperature after 30 min of laser irradiation (Figure 7). To determine the photothermal conversion efficiency (*η*) of gold nanostructures, the experimental data (ΔT versus time) were fitted to a mathematical model (Equation (1)) used by Roper [46].

Figure 8 shows *η* values determined for AuNR-CTAB and AuNR@SH-PECs. In accordance with this analysis (Figure 7), it can be observed that at the concentration of SH-PEC is 8.0 mg/mL, and the *η* value shows an apparent lower photothermal conversion capacity; this can be due to the increment of the polymeric-coated thickness deposited on the surface of the AuNR, which affects the absorption property of AuNR, probably due to the increase in scattered light intensity and diminishment of total absorbed light [52]. On the other hand, the *η* values calculated for AuNR@SH-PECs (1.0 to 4.0 mg/mL) showed negligible variations, and moreover, the *η* value remains practically equal to the *η* calculated for AuNR-CTAB.

### 3.8. Photothermal Effect of AuNR@SH-PEC on the Viability of Balb/c 3t3 and HeLa Cell Lines In Vitro Biological Evaluation

The photothermal effect of AuNR@SH-PECs on the viability of Balb/c 3t3 and HeLa and was evaluated by in vitro assays. The cytotoxic activity of AuNR@SH-PEC was assessed by conducting tests involving AuNR-CTAB and AuNR@SH-PECs at concentrations ranging from 6.25 × 10^9^ to 1.00 × 10^11^ AuNRs/mL, without subjecting them to laser irradiation. These results are shown in Figure 9. As was expected, the AuNR-CTAB greatly affected the cell viability of both cell lines, observing a survival percentage for Balb/c 3t3 and HeLa cells close to 10%. This is attributed to the high cytotoxicity of CTAB, since surfactants affect the mechanical properties of the cell membrane, restricting the use of AuNR-CTAB for biological purposes [53]. On the other hand, AuNR@SH-PEC slightly affected the viability of Balb/c 3t3, observing a survival of cells superior to 75% for the higher concentration of AuNR@SH-PEC tested (1.00 × 10^11^). On the other hand, the survival of the Balb/c 3t3 cells incubated in the presence of 6.25 × 10^9^ AuNR@SH-PEC/mL, for all formulations of AuNR@SH-PEC tested, was around 100%, suggesting that the replacing of CTAB for SH-PEC results in a biocompatible nanomaterial. Contrarily, HeLa cells showed greater susceptibility to the presence of AuNR@SH-PEC. As shown in Figure 9b, the cytotoxic effect of AuNR@SH-PEC on HeLa cells is influenced by both the concentration and specific formulation of AuNR@SH-PECs. In these regards, when the HeLa cells were incubated with AuNR@SH-PEC stabilized with 1 mg/mL of SH-PEC, the cell viability decreased from 97% to 70% across the evaluated AuNR@SH-PEC concentration range (6.25 × 10^9^ and 1 × 10^11^ AuNR@SH-PEC/mL). Furthermore, there was a noticeable increase in the toxicity of AuNR@SH-PECs to HeLa as the amount of SH-PEC used for the stabilization of gold nanorods increased. For instance, at a concentration of 6.25 × 10^9^ AuNR@SH-PEC/mL, the HeLa cell survival rate dropped from 97% to 54% when the cells were exposed to AuNR@SH-PEC stabilized with 8.0 mg/mL within the concentration range of 6.25 × 10^9^ to 1 × 10^11^. Based on these results, the concentration of 6.25 × 10^9^ of AuNR@SH-PEC/mL was chosen to assess the photothermal effects on the viability of both Balb/c 3t3 and HeLa cells. This decision was made to avoid potential toxicity arising from the type and concentration of AuNR@SH-PEC.

Then, the photothermal effect of gold nanorods on the viability of Balb/c 3t3 and Hela cells was assessed using a concentration of 6.25 × 10^9^ AuNR@SH-PEC/mL, irradiating with a CW laser (808 nm, 1 W). As depicted in Figure 10, both Balb/c 3t3 and HeLa cells subjected to laser irradiation exhibited a significant reduction in cell survival percentage. This outcome underscores the effectiveness of AuNR@SH-PEC in killing cells, supporting its potential as a photothermal system against cancer, in line with the proposition made earlier by Arellano-Galindo [54].

### 3.9. ROS Generation Assays

Reactive oxygen species (ROS) play a pivotal role in various cellular functions, including differentiation and the killing of cancerous cells [55,56]. In this context, the ROS production in Balb/c 3t3 and HeLa cells stimulated by the presence of AuNR@SH-PECs during the laser irradiation process was studied (Figure 11). In accordance with Figure 11, the irradiation of Balb/c 3t3 and HeLa cells with the 808 nm laser in the presence of AuNR@SH-PECs resulted in a slight increase in ROS production compared to the control cells (those laser-irradiated without AuNR@SH-PECs). Specifically, Balb/c 3t3 cells showed a 47% increase in ROS production when they were incubated with AuNR-@SH-PECs stabilized with 8 mg/mL. In contrast, HeLa cells reached peak ROS production with AuNR-@SH-PECs stabilized with 4 mg/mL. his differential response can be attributed to the heightened metabolic activity inherent to cancer cells [55,56]. These results suggest that AuNR@SH-PECs possess a great photothermal conversion capacity, but they cannot induce ROS production robustly.

## 4. Conclusions

In summary, our research culminated in the successful development of a novel nanoplatform, AuNR@SH-PEC. In accordance with the results, the methodology used to produce this nanomaterial is easy to develop and is replicable. Initially, a chemical modification of PEC was implemented to attach cysteamine molecules to the biopolymer structure and to introduce -SH groups, which have a high affinity for gold surfaces. This facilitated the substitution of CTAB, initially adsorbed onto the surface of AuNR, with SH-PEC molecules, resulting in a stable gold-based nanosystem. Importantly, the optical and photothermal properties of the synthesized AuNR@SH-PECs are comparable to those observed in AuNR-CTAB. Notably, the in vitro assays showed that AuNR@SH-PEC has superior biocompatibility compared to AuNR-CTAB. This consequently enhances its suitability for effortless integration into biological systems. The viability assays confirmed the potential of AuNRs@SH-PECs as a photothermal agent since the survival of HeLa cells was reduced to 40%. Based on these data, it is confidently stated that these nanoplatforms show significant potential for advancing the field of photothermal therapy, introducing a novel option in the fight against formidable diseases such as cancer.

## Figures and Tables

**Figure 1 pharmaceutics-15-02571-f001:**
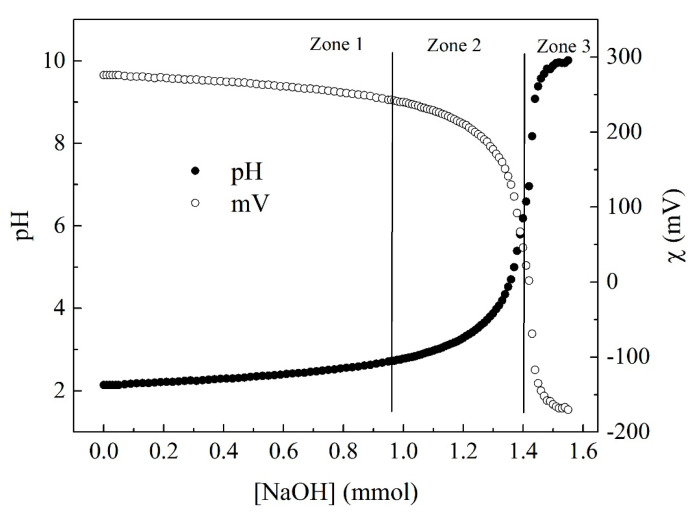
Potentiometric titration of PEC with NaOH.

**Figure 2 pharmaceutics-15-02571-f002:**
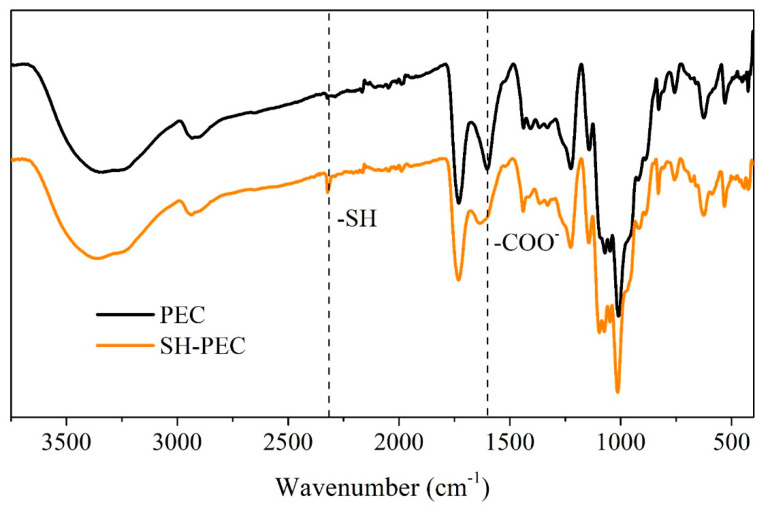
FTIR-ATR spectra recorded for PEC (black line) and SH-PEC (yellow line).

**Figure 3 pharmaceutics-15-02571-f003:**
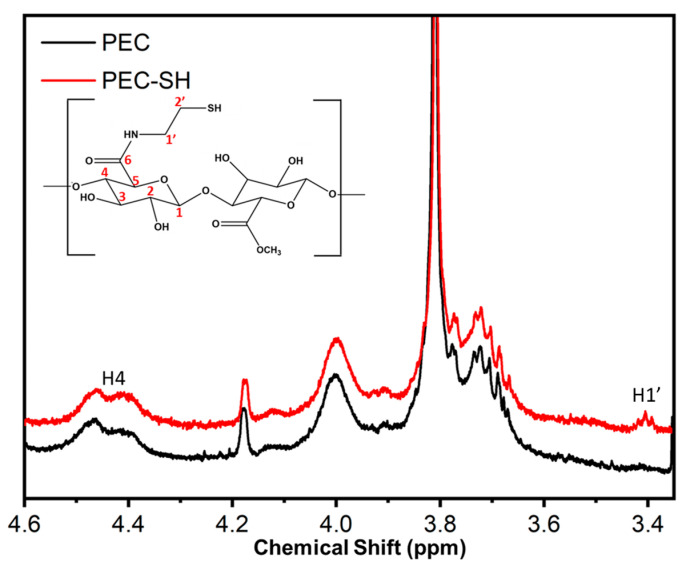
^1^H NMR spectra recorded for PEC (black line) and SH-PEC (red line).

**Figure 4 pharmaceutics-15-02571-f004:**
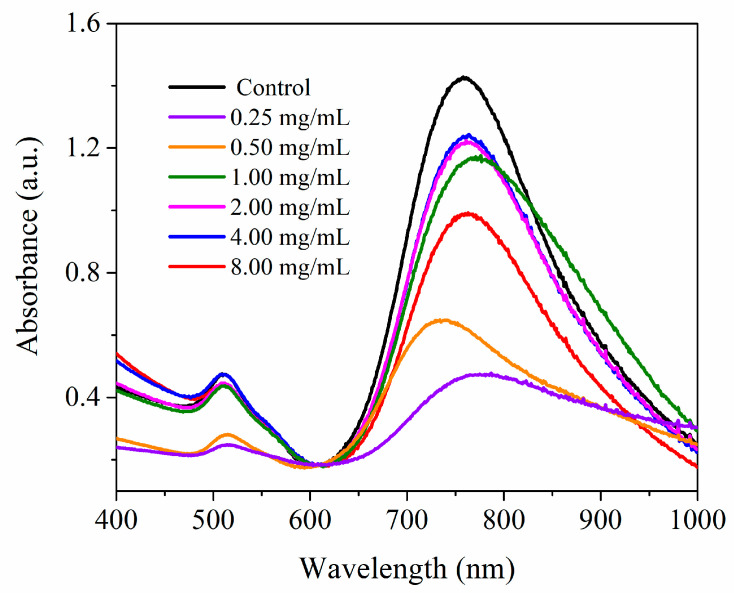
UV-Vis absorption spectra recorded for the AuNRs stabilized with CTAB and different amount of SH-PEC.

**Figure 5 pharmaceutics-15-02571-f005:**
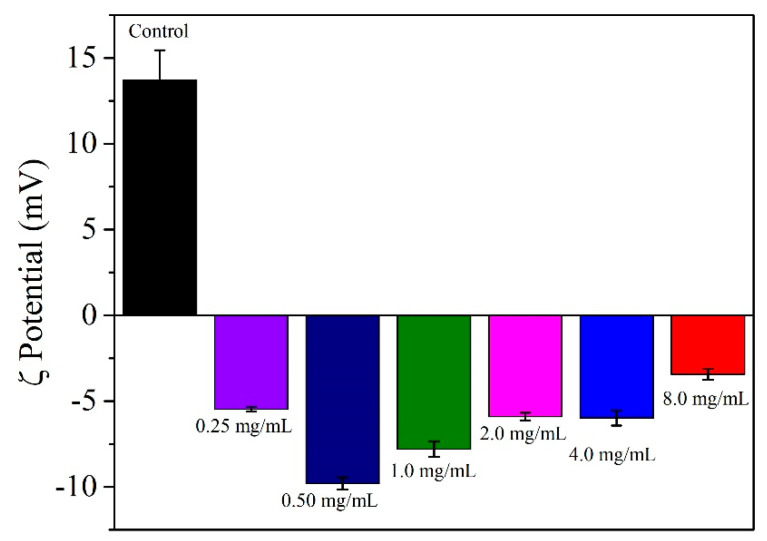
ζ potential recorded for AuNR stabilized with CTAB and for AuNR@SH-PEC in water.

**Figure 6 pharmaceutics-15-02571-f006:**
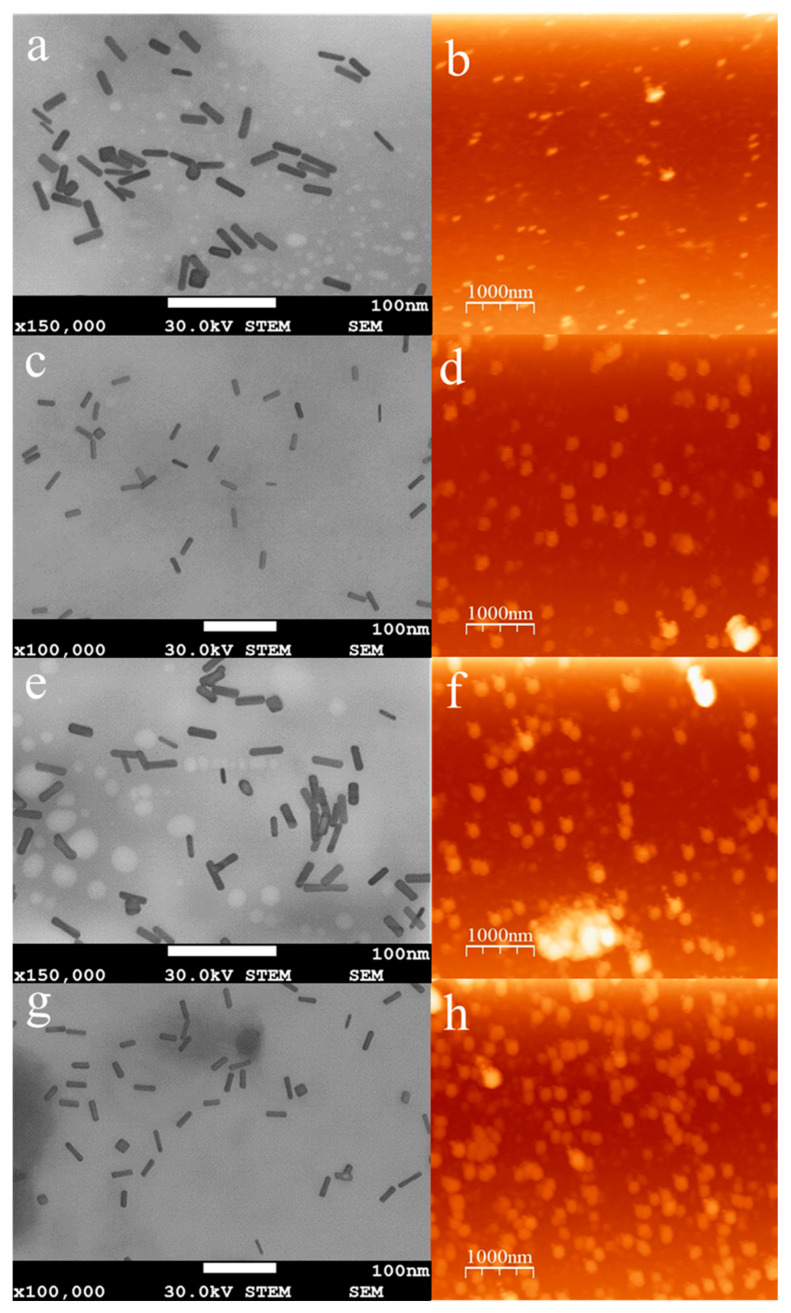
STEM and AFM images of AuNR@SH-PEC; (**a**,**e**) 8.0 mg/mL SH-PEC, (**b**,**f**) 4.0 mg/mL SH-PEC, (**c**,**g**) 2.0 mg/mL SH-PEC, (**d**,**h**) 1.0 mg/mL SH-PEC.

**Figure 7 pharmaceutics-15-02571-f007:**
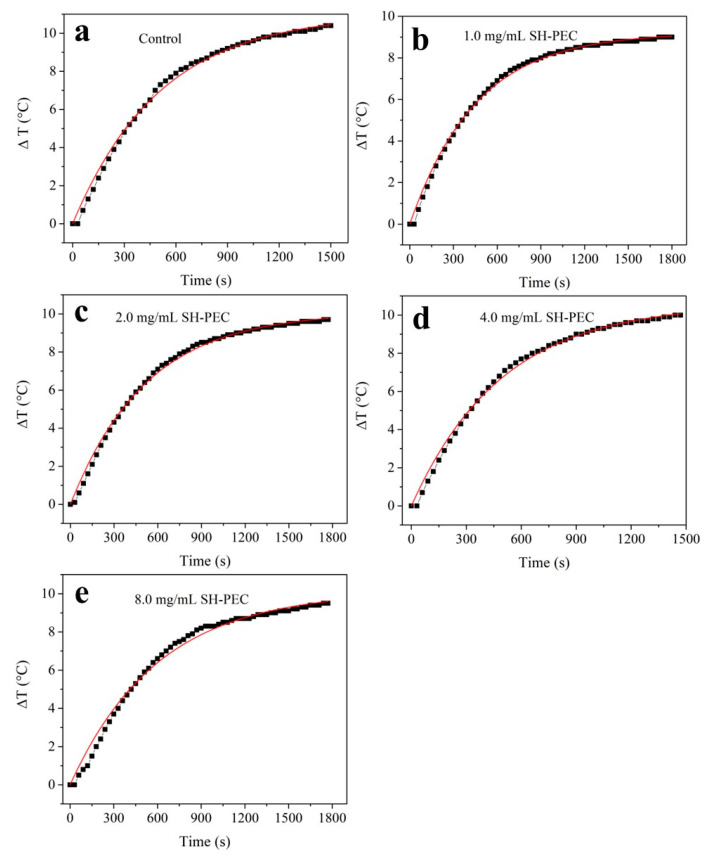
Photothermal conversion efficiency adjustments of AuNR@SH-PEC with different SH-PEC concentrations, (**a**) AuNRs control with CTAB, (**b**) 1.0 mg/mL, (**c**) 2.0 mg/mL, (**d**) 4.0 mg/mL, (**e**) 8.0 mg/mL.

**Figure 8 pharmaceutics-15-02571-f008:**
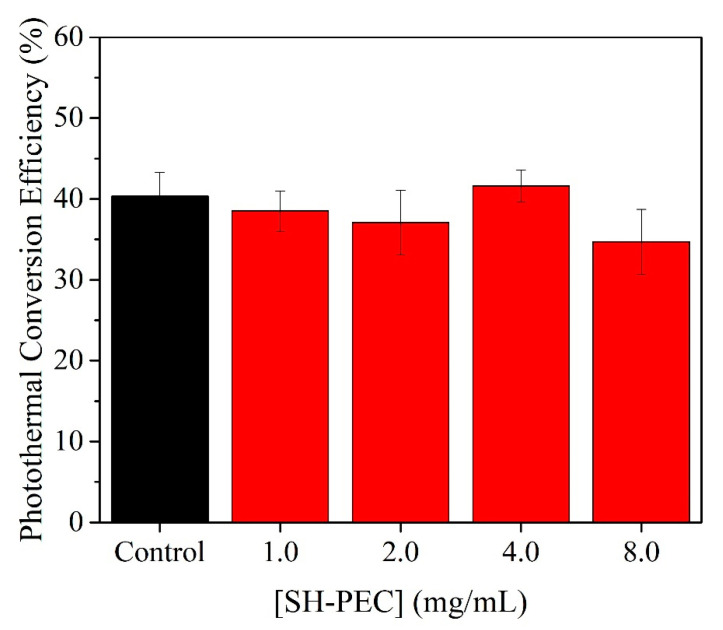
Photothermal efficiency (*η*) of AuNRs covered with different concentration of SH-PEC.

**Figure 9 pharmaceutics-15-02571-f009:**
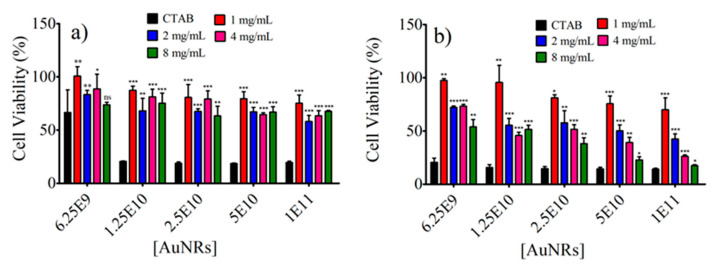
Cell viability assays of (**a**) fibroblasts Balb/c 3t3 cell line and (**b**) cervical cancer HeLa cell line in the presence of AuNR@SH-PEC irradiated with an 808 nm laser (1 W). *** *p* < 0.001; ** *p* < 0.05; * *p* < 0.05; ns indicates no significant difference vs. CTAB.

**Figure 10 pharmaceutics-15-02571-f010:**
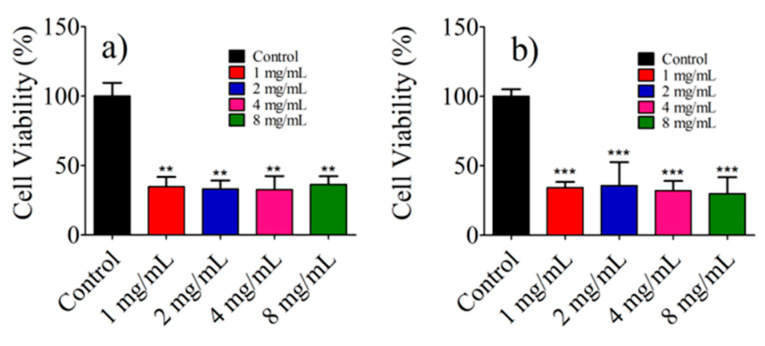
Cell viability assays of fibroblasts Balb/c 3t3 (**a**) and HeLa cells (**b**) treated with AuNR@SH-PEC (6.25 × 10^9^ AuNRs/mL) irradiated with 808 nm laser (1 W). *** *p* < 0.001; ** *p* < 0.05; ns indicates no significant difference vs. Control.

**Figure 11 pharmaceutics-15-02571-f011:**
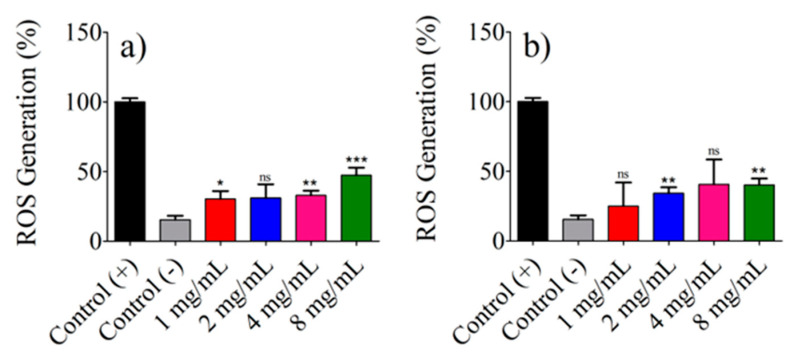
ROS production by (**a**) fibroblast Balb/c 3t3 cell line and (**b**) cervical cancer HeLa cell line irradiated with 808 nm laser (1 W) in presence of AuNR@SH-PEC. *** *p* < 0.001; ** *p* < 0.05; * *p* < 0.05; ns indicates no significant difference vs. Control (−).

**Table 1 pharmaceutics-15-02571-t001:** Hydrodynamic Diameter of AuNR@SH-PEC Obtained by DLS.

SH-PEC Concentration (mg/mL)	Hydrodynamic Diameter (nm)
0.25	177.0 ± 8.0
0.50	104.6 ± 19.0
1.00	50.8 ± 8.8
2.00	68.4 ± 6.4
4.00	114.2 ± 33.5
8.00	191.0 ± 8.6

## Data Availability

Appendix A.

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
