# Peer review of "Novel Gold Nanorods@Thiolated Pectin on the Killing of HeLa Cells by Photothermal Ablation"

_pharmaceutics, 2023, doi:10.3390/pharmaceutics15112571_

Round 1

Reviewer 1 Report

Comments and Suggestions for Authors

In this paper the authors present the synthesis of novel gold nanoparticles stabilized with thiolated pectin (SH-PEC) in replacement of the cytotoxic CTAB surfactant used for initial nanoparticles synthesis. The photothermal properties of the resulting NPs were evaluated by irradiating the aqueous suspension of nanoparticles in the NIR range. Moreover, the cell viability assays indicated the increased the biocompatibility of the nanoparticles, and the viability assays with irradiation showed a decrease in the cell viability in comparison to the cells without irradiation.

The results indicated that Au-NPs stabilized with thiolated pectin showed their potentiality in photothermal therapy. However, there are some aspect that should be revised and some additional studies should be done:

  1)      The authors claim that “there are no reports of AuNRs functionalized with PEC”. However there are some recent publications related to Au-nanoparticles coated with PEC and the study of their photothermal applications, especially for photodynamic therapy. The authors should improve the bibliography references and make a brief summarize of the closest examples in bibliography (i.e ACS Omega 2021, 6, 19, 12567–12576, Environmental Science and Pollution Research 2018, 25, 18476–18483 among others…)

  2)      It is recommended to improve the section Materials and Methods. The information concerning different techniques should be improved. For example, the characteristics of potentiometer (reference electrode, salt bridge, analyte and an indicator electrode). Instrument expecifications for FT-IR, the information about obtaining calibration curve in ICP-MS and reference standards used, etc…

  3)      There are some additional characterizations missing to have a close information about SH-PEC obtaining and purification. In fact there is not a characterization of SH-PEC purity. The authors should must to perform chemical analysis and NMR analysis to ensure the obtaining of the final ligand and the purity of final product. The raw data can be detailed in supplementary information.

  4)      DLS and z-potential spectra and recorded conditions should be showed in supplementary material.

  5)      In the same way ICP-MS of the final NPs should give information concerning the % of inorganic-organic part and could help to determine exactly the number of HS-PEC molecules per particle. The raw data can be detailed in supplementary information.

  6)      Could the authors correlate the UV-Vis bands shifts with size?. It would be interesting correlate the surface coating and possible aggregation processes with plasmon resonance.

  7)      Although production of ROS is not high, it can be considered the influence of increased ROS production (10-50% depending on the coating) in the cytotoxic effect of the nanoparticles. In fact, if we observe the cytotoxic effect in different cell lines there is a notable difference between fibroblast and HeLa cells that could be related to this effect since the ROS sensitivity of different cell lines can be dissimilar. Please, could the authors give some explanation in this sense?

Concerning the use of Pectin, it should be recommendable that the authors highlighted the biological implications of use Pectin, apart the benefits of being a biocompatible material (ref: International Journal of Biological Macromolecules, 185, 2021, 49-65)

8)      Other minor changes:

Last word in Abstract: change therapist by therapy.

In general terms, there is a notable piece of work to be published in Pharmaceutics. However, different aspects summarized in the present report should be addressed before acceptation for publication.

Author Response

Dear Reviewer

Thank you for your time and feedback. We have attached a PDF file containing our responses to all queries and suggestions.

Best regards

Reviewer 2 Report

Comments and Suggestions for Authors

1. Author should provide a discussion on the photostability of the synthesized AuNRs and whether the obtained nanoparticles (NPs) are photostable or not. Photostability is essential as it directly influences their reliability in therapeutic applications. Please refer to Journal of Alloys and Compounds, 2021, 876, 160175 and Nanomaterials, 2021, 11, 695.

2. There is a need for improvement in the manuscript's English language including grammar and typos.

3. The terminology "TRSP" should be corrected to "TSPR" throughout the manuscript.

Comments on the Quality of English Language

Can be improved

Author Response

(The authors gave the same response as above.)

Reviewer 3 Report

Comments and Suggestions for Authors

The main goal of the present paper is to develop easy and reproducible method to get a stable and biocompatible AuNR with promising application as photothermal agent. The authors used thiolated pectin (SH-PEC) to replace the CTAB, which is toxic. They demonstrated that SH-PEC is capable to replace the CTAB adsorbed on the surface of AuNRs. However, there some points to be analyzed:

1.Mine main concern is regarding the novelty of the paper, because the thiolated pectin (SH-PEC) have already used on gold nanoparticles, therefore the novelty will be limited to the shape (nanorods) what is not enough, see for example the following articles:

a) Artif Cells Nanomed Biotechnol

Shaivee Borker Varsha Pokharkar .

2018;46:826-835.

doi: 10.1080/21691401.2018.1470525.

Engineering of pectin-capped gold nanoparticles for delivery of doxorubicin to hepatocarcinoma cells: an insight into mechanism of cellular uptake

2. The removal of CTAB from the surface of Au Nps was demonstrated, but not from the solution. After removal form surface, CTAB should be not detectable by NMR spectroscopy, surface-enhanced Raman spectroscopy, or using pH dependent ζ-potential measurements. This test is esential because the use of PEC for removal of CTAB is the main aim of the present work.

3) The literature review is missing important works, for example the article mentioned in the point 2.

Comments on the Quality of English Language

I dont have comments

Author Response

Dear Reviewer

Thank you for your time and feedback. We have attached a PDF file containing our responses to all queries and suggestions

Best regards

Reviewer 4 Report

Comments and Suggestions for Authors

The manuscript describes the design of a new nanopartform based in AuNRs for application in cancer phototherapy. The replacement of CTAB as a AuNRs stabilizer is attempted with the synthetized thiolated PEC. The topic is of relevance, yet several considerations need to be addressed:

1. in the introduction, it is clear the need for new AuNRs stabilizers, however why PEC should be studied if there are already natural polysaccharides with proven stabilizing properties?

2. the last paragraph of the introduction is basically a sum up of the results, perhaps more appropriate in other sections (e.g., abstract)

3. the methodology section is crucial for the applicability of the new nanoplatform by the scientific community, yet is very scarce in details, namely:

a) section 2.2. how did the agitation was produced?

b) section 2.3 what is the volume of the dialysis and what type of tube was used

c) section 2.4, isn't a lyophilisation step missing?

d) section 2.5 what is the volume of the seeds?

4. in the results section the title should be revised to "results and discussion"

5. Figure 3 is difficult to distinguish the line of the control and the 0.5 mg/mL

6. section 3.6, please clarify what is meant by the "drying step" and how it may induce aggregation... isn't this sign of poor stability in the AuNRs@PEC?

7. Figure 10 and respective text, HeLa's data is similar to fibroblasts, please revise the text and add statistical analysis; also relate data with the obtained without irradiation

8.Figures 8 and 9 should be simpler and without so many sub-figures which make it more difficult to understand

There are several speeling issues to be addressed, for example

line 425 amend "determied" to "determined"

line 495, there are 2x "importantly"

please revise the whole manuscript and standardize the units, namely time

Author Response

(The authors gave the same response as above.)

Round 2

Reviewer 1 Report

Comments and Suggestions for Authors

I consider that most of the recommendations and comments have been addressed. However, I think authors could have made the effort to determine the metal content in the nanoconstruct by ICP-MS. The difficulty in treating the sample is not an adequate excuse for not having done it, especially if the sample preparation is a simple acid digestion. And the results using this technique sometimes give us more information than expected. I recommend that in future studies, authors also prioritize the determination of the metallic/inorganic content in these nanoplatforms using ICP-MS, as well as the more detailed study of aggregation using UV-Vis, especially if you look for a translation closer to clinical application.

Apart of this, in general I consider that the current manuscript is suitable for consider it for publication.

Reviewer 3 Report

Comments and Suggestions for Authors

The corrections improved properly the paper, the novelty is now clear

Comments on the Quality of English Language

no comments

Reviewer 4 Report

Comments and Suggestions for Authors

The authors have proceeded with the manuscript revision resulting in a significant improvement. It is recommended for publication.